# Self-Reported Non-Celiac Wheat Sensitivity and Other Food Sensitivities in Patients with Primary Sjögren’s Syndrome

**DOI:** 10.3390/nu17193172

**Published:** 2025-10-08

**Authors:** Aurelio Seidita, Pasquale Mansueto, Maurizio Soresi, Diana Di Liberto, Gabriele De Carlo, Gianluca Bisso, Salvatore Cosenza, Mirco Pistone, Alessandra Giuliano, Gabriele Spagnuolo, Clara Bertolino, Clarissa Bellanti, Roberto Citarrella, Lidia La Barbera, Giuliana Guggino, Antonio Carroccio

**Affiliations:** 1Unit of Internal Medicine, “V. Cervello” Hospital, Ospedali Riuniti “Villa Sofia-Cervello”, Via Trabucco, 180, 90146 Palermo, Italy; aurelio.seidita@unipa.it (A.S.); gianluca.bisso@virgilio.it (G.B.); alegiuliano94@gmail.com (A.G.); dr.clarabertolino@gmail.com (C.B.); 2Department of Health Promotion Sciences, Maternal and Infant Care, Internal Medicine and Medical Specialties (PROMISE), University of Palermo, 90127 Palermo, Italy; pasquale.mansueto@unipa.it (P.M.); maurizio.soresi@unipa.it (M.S.); roberto.citarrella@unipa.it (R.C.); lidia.labarbera@unipa.it (L.L.B.); giuliana.guggino@unipa.it (G.G.); 3Institute for Biomedical Research and Innovation (IRIB), National Research Council (CNR), 90146 Palermo, Italy; 4Department of Biomedicine, Neurosciences and Advanced Diagnostics (BIND), Institute of Biochemistry, University of Palermo, 90127 Palermo, Italy; diana.diliberto@unipa.it; 5Unit of Rheumatology, University Hospital of Palermo, 90127 Palermo, Italy

**Keywords:** primary Sjögren’s syndrome, non-celiac wheat sensitivity, multiple food sensitivity, wheat-free diet, intestinal permeability

## Abstract

Background: Wheat or cow’s milk intake might influence the primary Sjögren’s Syndrome (pSS) clinical manifestations. A high prevalence (20–30%) of autoimmune diseases, including pSS, has been reported in non-celiac wheat sensitivity (NCWS). This study aimed to identify the prevalence of self-reported NCWS and sensitivity/intolerance to other foods in patients with pSS, and to establish the specific clinical and immunological features of this subgroup of patients. Methods: 82 prospectively enrolled pSS patients were compared to 161 type 2 diabetes controls without rheumatological disease. The presence of a self-reported NCWS, and/or self-reported milk intolerance (SRMI), and/or multiple food sensitivity (MFS) was assessed by a validated questionnaire. Clinical and immunological features of pSS subjects, stratified according to the presence/absence of self-reported NCWS, were analyzed. Results: pSS patients had a higher frequency of self-reported NCWS (47.6% vs. 18.6%, *p* < 0.0001), SRMI (29.3% vs. 5.6%, *p* < 0.0001) and MFS (30.5% vs. 9.3% *p* < 0.0003) compared to controls. After the intake of wheat-containing products, 18 (21.9%) pSS patients reported the worsening of disease-specific symptoms, whereas 11 (13.4%) reported a significative clinical improvement after wheat-free diet (WFD) introduction. Moreover, 47.6% of pSS subjects complained of wheat-related gastrointestinal/extraintestinal disorders. No clinical/immunological feature differentiates pSS patients with and without self-reported NCWS, excluding a higher frequency of SRMI (39.5% vs. 11.9%, *p* = 0.01) and MFS (65.7% vs. 23.8%; *p* = 0.0004) in the former. Conclusions: This study shows a clear association between pSS and NCWS, confirming that wheat intake could be a common trigger of symptoms of both these conditions. WFD adoption seems to reduce both gastrointestinal/extraintestinal and pSS-specific symptoms in a subgroup of pSS patients, opening new possibilities for their clinical management.

## 1. Introduction

Sjögren’s Syndrome is one of the most frequent autoimmune rheumatological diseases, characterized by the typical lymphocytic infiltration of the lacrimal and salivary glands. It mainly manifests with keratoconjunctivitis sicca and xerostomia [1]. Its prevalence is estimated to range from 0.01% to 3.0% in the general population [2], mainly affecting middle-aged (sixth and seventh decade of life) women (female-to-male ratio: 9/1) [3,4,5,6,7]. It could be defined as primary Sjögren’s Syndrome (pSS), if it is not associated with other autoimmune disorders, or as secondary if it is diagnosed together with other systemic rheumatological diseases (e.g., systemic lupus erythematosus, systemic sclerosis or rheumatoid arthritis). Nevertheless, there is also a third option in which Sjögren’s Syndrome is described neither alone nor in the context of other systemic autoimmune disorders, but in association with organ-specific autoimmune diseases, such as celiac disease (CeD), primary biliary cholangitis, autoimmune hepatitis, autoimmune thyroid diseases, etc. [6,7,8].

Several biological processes are involved in pSS pathogenesis. In the initial phase, the innate immune response and CD4+ T cells play preponderant roles, while, in the later phases, both the innate immune system and the helper T cells of the adaptive immune system can induce B-cell activation and subsequently establish the positive feedback circle characterizing pSS [9,10]. A group of cytokines [e.g., Interleukin (IL)-1, IL-2, IL-6, Tumor Necrosis Factor (TNF)-α and Interferon-γ (IFN)-γ], produced by both immune cells and salivary gland epithelial cells, plays a main role in this process [11,12].

Since the first report of a ‘sicca syndrome’ in the early 1900s, by the Swedish physician Henrik Sjögren, and despite several physiopathological and clinical advances in understanding this condition [6,7,13], the precise mechanism underlying the deterioration of salivary and lacrimal gland function remains elusive. In addition, some authors have identified different subgroups of patients, characterized by various degrees of glandular and/or extraglandular inflammation and symptoms, that could probably benefit from differentiated therapeutic approaches [14]. Therefore, the current treatments for pSS (e.g., artificial saliva and secretagogues, such as pilocarpine and cevimeline) focus primarily on alleviating symptoms, as doubts persist regarding the advisability and the usefulness of attenuating or eliminating the inflammatory process [12,15,16].

Clinically, pSS is primarily characterized by ocular and/or mouth dryness (i.e., ‘sicca syndrome’). At least one of these symptoms affects 98% of pSS patients, with both occurring in 90% [5,6,7]. However, the disease generally progresses slowly over time [8,17]: there is an initial pre-clinical phase, characterized only by antibody positivity in the absence of clear signs and symptoms (pre-disease); there is a phase characterized only by glandular involvement with xerostomia and xerophthalmia (stage 1); there is a phase characterized by systemic extraglandular manifestations, including fatigue and confusion, but also neurological, musculoskeletal, and nephrological expressions, as well as the involvement of skin and other organs (stage 2); and there is a phase with a possible, though infrequent (about 5% [18]), evolution towards a lymphoid malignancy, usually non-Hodgkin’s lymphoma (stage 3).

In recent years, several research groups have highlighted how nutrition can influence the symptoms presented by patients with pSS, considering foods such as wheat or cow’s milk as possible triggers of this disease’s symptoms [19,20,21,22]. However, there are no univocal indications for a possible useful dietary approach in pSS patients. Some preliminary data have suggested that a gluten-free diet (GFD) may reduce sialadenitis and increase salivary flow in patients with coexisting pSS and CeD [22]. Furthermore, some patients with pSS experienced an inflammatory response [i.e., release of nitric oxide (NO)] following a rectal challenge with gluten [21].

In the last two decades, the existence of a wheat-related disorder in patients without CeD or IgE-mediated Wheat Allergy (WA) has been definitively established, being known as non-celiac wheat sensitivity (NCWS) [23]. Even if its exact prevalence in the general population is unknown, the prevalence of self-reported NCWS is approximately 10% [24].

Conflicting data have been reported on the underlying pathophysiology of this disease [25], but an increasing amount of evidence has shown that the intake of wheat components such as amylase-trypsin inhibitors (ATIs) might increase intestinal permeability (IP) and activate the innate immune response via the Toll-like receptor (TLR)-4 localized on myeloid cells, promoting the production of pro-inflammatory cytokines, which damage the intestinal mucosa [26]. Such a scenario likely increases the exposure of foods/microbial peptides to immune cells, leading to both local and systemic activation [26,27,28,29,30,31] and, finally, the onset of intestinal/extraintestinal clinical manifestations [23,25,30,32,33,34,35,36].

A high rate (20–30%) of autoimmune diseases has also been reported in NCWS patients. These include Hashimoto’s thyroiditis, undifferentiated connective tissue disease, psoriatic arthritis, seronegative spondylarthritis, and pSS [21,37,38,39].

In this scenario, it could be hypothesized that wheat intake might be one of the environmental factors that can worsen pSS symptoms. Therefore, the present study aimed to identify the prevalence of self-reported NCWS and sensitivity/intolerance to other foods in a population of patients with pSS, and to establish, at the same time, the specific clinical and immunological features of this subgroup of patients compared to subjects with pSS who do not report concomitant food sensitivity.

## 2. Materials and Methods

This was a prospective, single-center study conducted on patients suffering from pSS, diagnosed according to the American College of Rheumatology/European League Against Rheumatism (ACR/EULAR) criteria [40], who consecutively enrolled at the Sjögren’s Disease Outpatient Clinic of the Rheumatology Unit of the University Hospital of Palermo. Age- and sex-matched subjects with type 2 diabetes mellitus (DM) from the “Cardiovascular Risk Prevention Clinic” of the Internal Medicine Unit of the same hospital were recruited as a control group. These patients were chosen as controls because DM is, by definition, a metabolic-related disease with a minimal or absent autoimmunity component [41,42]. Data collected from medical records and interviews were uploaded into a computerized database and subsequently analyzed.

### 2.1. Inclusion and Exclusion

Inclusion criteria for pSS patients
Age ≥18 and ≤75 years;Diagnosis of pSS according to the 2016 ACR/EULAR criteria [40];Written consent to participate in the study.

Inclusion criteria for DM patients
Age ≥18 and ≤75 years;Absence of the minimal criteria for pSS [40], or no diagnosis of other autoimmune disorders (excluding Hashimoto’s thyroiditis), in accordance with the current international criteria;Written consent to participate in the study.

Exclusion criteria
Diagnosis of CeD, according to current diagnostic criteria [43];Self-exclusion of wheat from the diet and refusal to reintroduce it for diagnostic purposes before entering the study (whenever required to exclude CeD);Drug and/or alcohol abuse (>30 g/day for men and >20 g/day for women);Treatment with systemic steroids and/or non-steroidal anti-inflammatory and/or immunosuppressive drugs in the 2 weeks before duodenal biopsy (whenever required to exclude CeD);Pregnancy or breastfeeding;Diagnosis of chronic inflammatory bowel disease or other organic pathologies affecting the digestive system (e.g., WA, microscopic colitis, diverticulitis, segmental colitis associated with diverticulosis, etc.), neurological diseases, major psychiatric disorders, infectious diseases, immunological deficiencies, and impairment limiting physical activity;Incomplete medical records;Lack of a clinical follow-up of at least 12 months after diagnosis with >2 outpatient visits during the follow-up period.

### 2.2. Outcomes

Primary outcome: prevalence of self-reported NCWS and/or other food sensitivities in pSS patients

To evaluate the presence of a self-reported NCWS and/or a self-reported sensitivity to other foods [i.e., self-reported milk intolerance (SRMI), defined as a self-reported clinical reaction to milk-containing foods independently of lactose breath test positivity, and multiple food sensitivity (MFS), defined as a self-reported clinical reaction to foods different from gluten/wheat and milk and dairy products] in patients with pSS, an ad hoc-designed questionnaire was administered. This was a modified version of a questionnaire already validated in previous studies [44] and consisted of 3 sections investigating different aspects of both the demographic features and the clinical manifestations of the patients (see Appendix A). Any modification in symptoms perceived as related to food intake was described using both a specific question and a visual analog scale.
Secondary outcome: evaluation of the demographic, clinical and immunological characteristics of pSS patients with or without self-reported NCWS

All patients in the study group were analyzed for pSS by their clinical and immunological features. They were then stratified based on the presence or absence of a self-reported NCWS, and finally compared to identify any characteristics that could differentiate the two resulting subpopulations.

The following standardized and validated international scores were used:The EULAR SS patient-reported index (ESSPRI) [45];The EULAR SS disease activity index (ESSDAI) [46].

To assess some of the main immunological features of pSS, the following assays were performed:-Anti-nuclear antibody (ANA) titers and patterns;-Anti-SSA/Ro antibodies (Sjögren’s-syndrome-related antigen A);-Anti-SSB/La antibodies (Sjögren’s-syndrome-related antigen B);-Rheumatoid factor (RF);-C3 and C4 complement component;-γ-globulinemia;-Monoclonal component of the γ-globulin fraction.

To reduce inter- and intra-laboratory variability, all assays were performed by the same physician at the laboratories of the University Hospital of Palermo, using specific commercial kits and strictly following the manufacturers’ instructions.

### 2.3. Statistical Analysis

Data were expressed as mean ± standard deviation when the distribution was Gaussian and Student’s *t* test was used to evaluate differences in means between groups. Otherwise, data were expressed as median and interquartile range (IQR) values and analyzed with the Kruskal–Wallis and Mann–Whitney U tests. The χ^2^ test and Fisher’s exact test were used to compare values of frequency in the various population groups.

Multiple logistic regression analysis was performed to estimate the independence of the association between pSS and a diagnosis of self-reported NCWS, SRMI and MFS.

The SPSS Statistics, version 27.0 (Chicago, IL, USA), and MedCalc, version 22.0 (Acacialaan, Ostend, Belgium), software were used for the statistical analysis.

All subjects agreed to participate in the study and provided written informed consent. The protocol was approved by the Ethics Committee of the University Hospital of Palermo (report no. 11/2022) and the study was registered on the CliniaclTrials.gov website (protocol no. NCT05644795).

## 3. Results

The medical records of 127 patients suffering from pSS were analyzed together with those of 283 DM control patients. After the application of the inclusion and exclusion criteria, 82 patients with pSS and 161 control patients were eligible and gave their consent to participate in the study (Appendix A).

Table 1 and Table 2 show the main demographic, sociological, and clinical features of the pSS and control subjects (additional clinical and immunological characteristics of the pSS subjects are reported in Appendix A). All pSS subjects were Caucasian (*p* = 0.03), usually had a higher qualification (higher number of subjects who had attended high school or university, *p* < 0.0001 and *p* = 0.025, respectively), and were in more stable employment (lower prevalence of unemployed, *p* = 0.022) than controls. The former, compared to controls, more frequently complained of abdominal pain/discomfort (68.3% vs. 42.9%, *p* = 0.0003), usually recurring more than 10 days per month (*p* = 0.0021); abdominal swelling (60.9% vs. 24.2%, *p* < 0.0001); changes in usual bowel moments (*p* = 0.049); as well as a correlation between intestinal disorder onset and episodes of psychophysical stress (*p* = 0.049). They also reported more comorbidities, such as neuropsychiatric symptoms (anxiety, *p* = 0.0008, and chronic headache, *p* = 0.001), thyroid disease (*p* < 0.0001), other non-systemic autoimmune diseases (*p* < 0.0001), and fibromyalgia *p* < 0.0001).

### 3.1. Primary Outcome: Prevalence of Self-Reported NCWS and/or Sensitivities/Intolerance to Other Foods

Figure 1 shows that patients with pSS had a higher frequency of self-reported NCWS (*p* < 0.0001), SRMI (*p* < 0.0001) and MFS (*p* < 0.0003) compared to controls.

Table 3 reports the symptoms referred to by both pSS patients and controls after eating wheat-containing products. Overall, 21.9% (N = 18) of pSS patients reported a worsening of pSS-specific symptoms after the intake of wheat-containing products, the most frequent being xerostomia, either alone (44.4%) or in association with xerophthalmia (44.4%).

Among these wheat-sensitive pSS subjects, 11 (61.1%) reported an improvement in their specific pSS symptoms after removing wheat from their diet, which was satisfactory (N = 5, 45.4%) or even permitted a therapy reduction (N = 4, 36.4%) in more than 80%. In addition, out of 82 patients with pSS, 39 (47.6%) complained of symptoms other than pSS-related ones after eating wheat-containing products thatwere primarily irritable bowel syndrome (IBS)-like [such as abdominal pain (*p* = 0.002), bowel movement disorders (diarrhea, *p* = 0.028), and flatulence (*p* < 0.0001)], but also extraintestinal and constitutional ones, such as asthenia (*p* = 0.027), headache (*p* = 0.006), numbness or a prickling sensation on the skin (*p* = 0.049), and joint pain (*p* = 0.045).

Similar symptoms were also reported by control patients, but with a significantly lower prevalence (18.6%, *p* < 0.0001). Analyzing the clinical features of the wheat-sensitive pSS and wheat-sensitive control subjects, the former complained of a long-standing history of wheat-dependent symptoms (median 120 months), often recurring daily and persisting longer, triggered by any type of wheat-containing foods (unlike the controls, who complained of symptoms almost exclusively after eating pasta or pizza).

Approximately 60% of patients in both groups had consulted a physician for these disorders, but in the pSS patients this was more frequently a gastroenterologist (*p* = 0.021), resulting in the prescription of CeD tests, while controls mainly consulted an allergist, resulting in the prescription of prick and patch tests. ‘No explanation’ for the reported symptoms was the ‘final diagnosis’ most frequently given by specialists, especially in the control subjects (*p* = 0.004). Finally, although 30 subjects in the control group complained of symptoms after eating wheat-containing foods, none had undergone a period of WFD, whereas more than 50% of the wheat-sensitive pSS patients had tried wheat elimination (*p* < 0.0001), with relief of one or more of the symptoms in 95% of cases.

Table 4 shows the results of the questionnaire on self-reported intolerance to non-wheat-containing foods in pSS patients and controls (details on the foods identified are reported in Appendix A). Similarly to what was reported for wheat-containing foods, pSS patients reported both pSS and non-pSS symptoms related to food intake. More specifically, 10 pSS patients (12.2%) complained of worsening xerostomia, either alone (50%) or in association with xeropthalmia (50%), and 80.0% of them reported symptom improvement after excluding non-wheat-containing foods from their diet, which was at least satisfactory in more than 60% of cases. A statistically significant difference was shown between the pSS and control patients for the onset of non-pSS-related symptoms (42.7% vs. 13.0%, *p* < 0.0001). Both groups complained of intestinal and extraintestinal symptoms, but diarrhea, belching and flatulence were more frequently reported by pSS subjects (*p* = 0.01, 0.027, and 0.007, respectively). Symptoms almost always occurred after a specific food was eaten in the controls (*p* = 0.045), who, however, less frequently consulted a physician compared to pSS patients (*p* < 0.0001).

Finally, multiple logistic regression analysis confirmed the association between pSS and self-reported NCWS as well as SRMI and MFS (Table 5).

### 3.2. Secondary Outcome: Evaluation of the Demographic, Clinical and Immunological Features of pSS Patients With or Without Self-Reported NCWS

None of the demographic, clinical, immunological and therapeutic features typical of pSS were significantly different between pSS patients with (n = 39) or without (n = 43) self-reported NCWS, although the frequency of both SRMI (39.5% vs. 11.9%, *p* = 0.01) and MFS (65.7% vs. 23.8%; *p* = 0.0004) was higher in the former (Appendix A). Multiple logistic regression analysis confirmed the association between the presence of MFS and self-reported NCWS (RR 2.76, 95% CI 1.53–4.98, *p* = 0.0008).

## 4. Discussion

The clinical management of patients with pSS has not yet found unanimous consensus in the scientific community [12,15,16], likely because this condition presents clusters of patients with different inflammatory/immune and clinical features [14]. To date, therapy is primarily based on the use of lubricant ointments, local immunosuppressants, artificial saliva and, in cases of severe extraglandular involvement, systemic immunosuppressants, such as glucocorticoids, methotrexate, azathioprine, hydroxychloroquine, etc. Sometimes, it even involves the use of anti-CD20 B lymphocyte-depleting agents (i.e., rituximab) [7,47]. More recently, new therapeutic perspectives have been applied in patients affected by pSS, including low doses of IL-2 (able to modulate immune cells response enhancing Tregs and suppressing the pro-inflammatory subsets) [48,49], nipocalimab (a monoclonal antibody that reduces circulating IgG levels by selectively blocking their interactions with the neonatal Fc receptor) [50,51], and ianalumab (a B cell-activating factor receptor inhibitor and B-cell depleter monoclonal antibody) [52].

In recent years, numerous research groups have focused their attention on how diet can influence inflammation and symptoms in patients with pSS [53,54]; however, to date, there is no univocal consensus about the dietary approach that a pSS patient should follow. Nevertheless, the role of nutrition is becoming fundamental as a consequence of the increasingly personalized health care approach, so dietary interventions and the possible use of food supplements for patients suffering from pSS need to be appropriately analyzed to identify new therapeutic and preventive strategies [19]. In this scenario, it seems that adherence to the Mediterranean diet can positively influence both the clinical manifestations and degree of inflammation of pSS [53]. Other studies, however, have hypothesized that wheat may have a potential pathogenetic role in pSS and that introducing a GFD can improve symptoms in patients with this condition [55].

CeD patients with pSS following a rigorous GFD had less sialadenitis and higher salivary flow rates compared to those on a gluten-containing diet [22]. In addition, patients with pSS showed increased sensitivity to rectally administered gluten [21].

These findings may suggest that gluten could represent an environmental risk factor for the onset of pSS symptoms; thus, a GFD could be advocated as a potential therapeutic approach for pSS patients. Nevertheless, a GFD is often burdened by nutritional deficiencies, even serious ones, and should never be started without specific medical prescription [56]. Consequently, to date, the role of GFD in pSS still needs to be studied and this diet cannot be recommended on the basis of current scientific evidence.

In addition to wheat, other foods have been analyzed as potential environmental triggers for pSS symptoms, including cow’s milk proteins [20].

The apparent high propensity of pSS patients towards some kind of “sensitivity to food”, including wheat/gluten and milk, may support the hypothesis of a “non-IgE-mediated sensitivity”, whic has already been advocated to explain the physiopathological processes underlying conditions such as NCWS [57] and cow’s milk protein allergy [58,59].

In addition, some studies have reported a high rate of autoimmune disorders, including pSS, among patients suffering from NCWS (often associated with MFS) [38,39], thus reinforcing the hypothesis that dietary triggers could be a common pathogenetic factor shared by these two conditions. However, the results obtained suggest that further research is required into the possible impact of adverse food reactions on pSS symptoms and disease activity.

In view of this evidence, the main aim of our study was to identify the prevalence and clinical features of self-reported NCWS and/or sensitivity to foods other than gluten/wheat-containing ones (e.g., SRMI and MFS) in a population of pSS patients, comparing them with those found in a control population with a minimal or absent component of autoimmunity (DM patients) [41,42].

Analysis of the clinical and immunological features of our pSS study population showed that the disease was under good control in most of the patients enrolled, with ESSDAI and ESSPRI values classifying them at a low or at most moderate degree of disease activity, both locally and systemically. Moreover, the immunological characteristics of our patients were perfectly comparable to those reported in international scientific literature [60].

The comparison between the demographic and sociological features of the two groups showed a higher educational status and a more stable employment level with a higher income in the pSS subjects. These data are consistent with the known epidemiology of the two diseases [61,62,63].

Overall, the patients with pSS reported the onset of episodes of abdominal pain, heaviness, discomfort, or swelling in the previous 12 months more often and with a higher frequency than controls, all conditions that are frequently associated with an alteration in bowel movements and episodes of psychophysical stress.

Following a more in-depth analysis, based on the use of an already validated questionnaire for the identification of intestinal and extraintestinal disorders related to the intake of foods (either containing or not containing wheat flour) [44], we found that these intestinal symptoms were associated with the intake of specific foods more often in the pSS subjects than in controls. In our study, almost 50% of pSS patients self-reported NCWS, characterized by intestinal IBS-like (e.g., abdominal pain, diarrhea and flatulence) and extraintestinal (e.g., asthenia, headache, numbness or prickling sensation on the skin, and joint pain) symptoms more frequently than controls. Similarly, almost 30% of pSS patients referred to SRMI and MFS, characterized by diarrhea, belching and flatulence, more frequently than controls. In spite of there being few reports in the literature of such a high prevalence of IBS-like symptoms in patients with pSS, our result should encourage researchers to include intestinal manifestations as one of the characterizing features of pSS, especially if they are related to a food sensitivity condition [64]. Supporting our findings, up to 95% of pSS patients in a French study complained of abdominal symptoms, especially pain, discomfort, tension and constipation, but no specific correlation with food intake was reported [65]. In addition, in another study in 2017, a 2.6% frequency of pSS was found among patients with IBS [66].

Comparing the prevalence of self-reported NCWS in the two populations of our study (47.6% vs. 18.6%), the data results from controls were virtually identical to those reported in the general population according to many international studies [44,67,68,69,70,71,72,73], whereas they were very significantly higher in patients with pSS. Furthermore, in the pSS patients, many of the comorbidities referred to by patients with self-reported NCWS, such as neuropsychiatric symptoms (anxiety and chronic headache), thyroid disease, other autoimmune diseases, and fibromyalgia, as well as SRMI and MFS, were found more frequently than in the control group [23,57,74,75,76]. All this seems to reinforce what has already been previously reported in various studies, including some by our research group: a close association between patients with NCWS, diagnosed using international criteria [75], and autoimmune diseases, including pSS [21,37,38,39]. This latter evidence was further strengthened by the multiple logistic regression analysis of our study, which proved the existence of an association between pSS diagnosis and the self-reported NCWS, SRMI and MFS conditions.

A potentially even more relevant result, but one which definitely requires further targeted in-depth analysis, is the reported association between typical pSS symptoms (i.e., xerostomia and xerophthalmia) and self-reported NCWS, SRMI, and MFS. Approximately 22% and 12% of pSS patients complained of a worsening of disease-specific symptoms related to the intake of wheat-containing or non-wheat-containing foods, respectively. Moreover, a large percentage of these subjects referred to a significant improvement in these symptoms on eliminating the indicated foods from their diet (61.1% among self-reported NCWS and 80.0% among SRMI/MFS). To our knowledge, very few studies have analyzed the role of a wheat-free diet (WFD) in pSS patients, demonstrating its putative role in the clinical management of these subjects [20,21,22]. Among these studies, a very recent paper hypothesized that a GFD in association with K^+^ rich diet, may contribute to xerophthalmia and/or xerostomia relief in selected pSS patients with certified low blood levels of potassium. However, researchers only advanced this hypothesis based on indirect data from other studies, some of which were not specifically conducted on patients with pSS, without providing original data to support their hypothesis [77].

Our study, although preliminary and based on questionnaires, is the first to report such a close association between pSS and NCWS. In this scenario, the possible physiopathological link could be an alteration in intestinal permeability (IP). Exposure to wheat would cause the release of zonulin, which, by binding to the surface of intestinal epithelial cells, would modify the cellular cytoskeleton and cause the loss of the normal function of occludins, ultimately leading to an increase in IP [78]. This would result in increased exposure of the immune system cells of the lamina propria to wheat molecules (e.g., gliadin, ATIs, etc.), with an increased infiltration and the activation of myeloid and dendritic cells in the intestinal mucosa and lymph nodes. The activation of the innate immune response, via TLR-4 localized on myeloid cells, could promote the production of pro-inflammatory cytokines, which would damage the intestinal mucosa [26,79], further increasing the exposure of foods/microbial peptides to immune cells [26,27,28,29,30,31]. This local inflammatory response would have systemic repercussions, with an alteration of the normal cytokine pattern and infiltration of monocytes/macrophages and T cells into the salivary and lacrimal glands, thus contributing to the onset or exacerbation of the clinical manifestations of pSS as an environmental factor. Obviously, to date, this is only a hypothesis based on indirect evidence from studies not specifically aimed at investigating this association, and we hope to confirm it as soon as possible. The present results, in fact, are only the preliminary part of a larger prospective study aimed at confirming NCWS diagnosis in subjects with pSS by a double-blind, placebo-controlled wheat challenge (DBPCWC), and analyzing, at the same time, immunological and intestinal permeability patterns related to wheat exposure in these patients.

A further interesting finding that emerged in our study was the difference in the specialist consultations required for the in-depth analysis of the onset of symptoms after wheat-containing food intake in the self-reported NCWS groups. In the pSS patient group, we mainly consulted a gastroenterologist as a specialist, and among the possible diagnoses proposed, CeD, IBS (so much so that 40.2% of patients with pSS reported having been diagnosed with IBS), and NCWS emerged; some of these patients underwent a WFD trial and subsequently reported symptom relief. In contrast, in the control group, the main specialist involved was an allergist, and only a very small portion of these patients were given a diagnostic hypothesis of CeD, IBS or NCWS, resulting in a greater number of patients leaving without a possible explanation. This difference may be due to the lower frequency of symptoms or to a greater tolerance in the control group, as well as to the possible socio-cultural differences in the subgroups, or even to the possible ‘prejudice’ that DM subjects are unlikely to present gastrointestinal disorders related to the intake of starchy foods.

As regards the secondary outcome of our research, none of the demographic, clinical (including the ESSPRI and the ESSDAI disease activity score), immunological and therapeutic features analyzed showed statistically significant differences between the ‘wheat sensitive’ and ‘non-wheat sensitive’ pSS subjects, with the exception of SRMI and MFS, which had a higher prevalence in patients who self-reported NCWS compared to those who did not. This result seems to be in perfect agreement with other studies conducted in NCWS patients, in which a high percentage presented MFS and even more often complained of SRMI. For example, in a study by our group, of 276 NCWS patients diagnosed with DBPCWC, 74.6% complained of MFS [23], which was such a high percentage that we considered that at least a subgroup of NCWS subjects might have a form of non-IgE-mediated food allergy [57]. This hypothesis also seems to be in line with other studies specifically conducted in pSS patients, which have shown a high prevalence of food sensitivities [64], thus representing a further physiopathological link between pSS and NCWS.

Our study has limitations that should be highlighted. The limited number of patients recruited, both in the pSS and in the control group, may have resulted in a beta error and prevented some of the parameters examined from reaching statistical significance. The NCWS, SRMI, and MFS diagnoses described in our study are conditions self-reported by the patients and not confirmed by a diagnostic gold standard test, which is the double-blind placebo-controlled challenge, including gluten/wheat or any other food self-reported as a symptom trigger. Therefore, the data obtained must be considered with extreme caution. In addition, in this phase of the study, no specific analyses of IP and inflammatory biomarkers have yet been carried out. Thus, to date, the existence of a physiopathological link between the two conditions still remains strictly confined to the realm of hypotheses and further studies (already planned) are necessary to confirm or deny it.

However, although preliminary and based exclusively on questionnaires for the identification of a self-perceived condition, this is the first study to have systematically analyzed patients with pSS, diagnosed according to international criteria, and to have found evidence for an association between the intake of specific foods and the onset of both pSS-specific and non-pSS-specific symptoms. Moreover, the pSS subgroup was specifically enrolled at a rheumatological outpatient clinic, thus avoiding the obvious selection bias that usually occurs when patients are enrolled in outpatient clinics specialized in gastroenterological/food-related disorders.

## 5. Conclusions

Starting from the hypothesis of a potential pathogenetic role of gluten/wheat, as well as other alimentary triggers, in pSS and other autoimmune diseases, our study showed the presence of a statistically significant higher prevalence of self-reported NCWS, SRMI and MFS in patients with pSS (47.6%, 29.3% and 30.5%, respectively) compared to a control group without autoimmune diseases (18.6%, 5.6% and 9.3%, respectively). These patients complained of both IBS-like intestinal and extraintestinal symptoms, which seemed to worsen following the intake of specific trigger foods and improved after their removal from the diet. Of note, a large percentage of these subjects also reported a worsening of pSS-specific symptoms and their improvement upon following an elimination diet. This evidence seems to confirm that exposure to food antigens and IP impairment can be important pathophysiological mechanisms in the genesis of both pSS and NCWS, and that, consequently, a dietary approach can be a valid alternative in the clinical management of these patients.

Nevertheless, the evaluation of the demographic, clinical and immunological features of pSS patients with self-reported NCWS compared to those not self-reporting the disorder did not identify any specific characteristics that could differentiate the two groups, except for the greater frequency of association with SRMI and MFS.

This study represents a first step in the identification and definition of wheat sensitivity in patients suffering from pSS. An in-depth prospective study, specifically designed to evaluate the biomolecular mechanisms underlying this association (i.e., IP and inflammatory processes), is essential to understand if wheat can be identified as an environmental factor able to induce symptoms, at least in a subgroup of patients with pSS, thus supporting a complementary dietary approach in the treatment of the disease.

## Figures and Tables

**Figure 1 nutrients-17-03172-f001:**
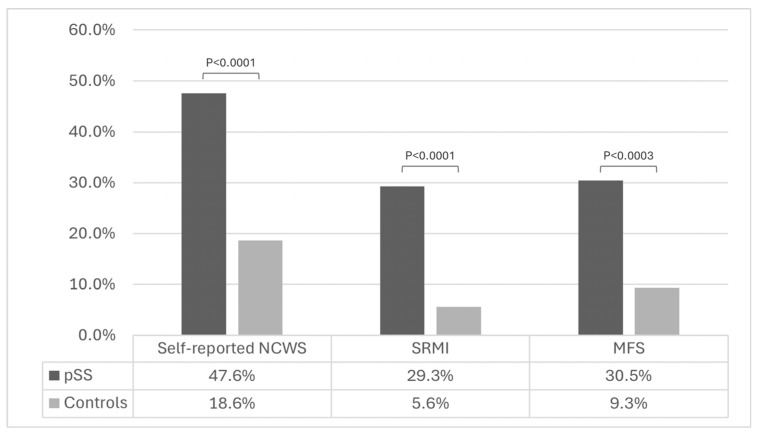
Prevalence of self-reported non-celiac wheat sensitivity (NCWS), self-reported milk intolerance (SRMI), and multiple food sensitivity (MFS) in primary Sjogren Syndrome (pSS) and control patients.

**Table 1 nutrients-17-03172-t001:** Demographic and sociological features of pSS patients and controls.

	pSS (N = 82)	Controls (N = 161)	*p*
Age (years) (mean ± SD)	62.5 ± 11.1	63.8 ± 12.7	NS
Gender (n, %)			
1. Male	4 (4.9)	20 (12.4)	NS
2. Female	78 (91.0)	141 (87.6)	
Ethnicity (n, %)			
1. Caucasian	82 (100.0)	146 (90.7)	0.03
2. African	0 (0.0)	15 (9.3)	NS
3. Asian	0 (0.0)	0 (0.0)	NS
4. Middle Eastern	0 (0.0)	0 (0.0)	NS
Marital status (n, %)			
1. Single	11 (13.4)	12 (7.5)	NS
2. Married	58 (70.7)	125 (77.6)	NS
3. Divorced	3 (3.7)	9 (5.6)	NS
4. Widowed	10 (12.2)	15 (9.3)	NS
Qualifications (n, %)			
1. None	3 (3.7)	6 (3.7)	NS
2. Elementary school diploma	8 (9.8)	51 (31.7)	0.0003
3. Middle school diploma	32 (39.0)	86 (53.4)	0.047
4. High school diploma	29 (35.4)	12 (7.5)	<0.0001
5. University degree	10 (12.2)	6 (3.7)	0.025
Employment status (n, %)			
1. Employee	16 (19.5)	36 (22.4)	NS
2. Freelance/Professional	5 (6.1)	6 (3.7)	NS
3. Laborer/unskilled worker, craftsperson	2 (2.4)	0 (0.0)	NS
4. Unemployed	43 (52.5)	110 (68.3)	0.022
5. Looking for work	2 (2.4)	3 (1.9)	NS
6. Unable to work	0 (0.0)	0 (0.0)	NS
7. Other	14 (17.1)	6 (3.7)	0.0009

NS = not significant; pSS = primary Sjögren’s Syndrome.

**Table 2 nutrients-17-03172-t002:** Gastrointestinal symptoms and main comorbidities of pSS patients and controls.

	pSS (N = 82)	Controls (N = 161)	*p*
Episodes of abdominal pain or heaviness or discomfort in the last 12 months (n, %)	56 (68.3)	69 (42.9)	0.0003
Frequency of episodes of abdominal pain or heaviness or discomfort (n, %)	N = 56	N = 69	
1. One day a month	9 (16.1)	9 (13)	NS
2. Two days a month	5 (8.9)	12 (17.4)	NS
3. Three days a month	8 (14.3)	12 (17.4)	NS
4. Four days a month	7 (12.5)	15 (21.7)	NS
5. 5–10 days per month	9 (16.1)	15 (21.7)	NS
6. More than 10 days a month	18 (32.1)	6 (8.7)	0.0021
Reduction in abdominal pain or heaviness or discomfort after having a bowel movement (n, %)	N = 56	N = 69	
	41 (73.2)	48 (69.6)	NS
Abdominal swelling in the last 12 months (n, %)	50 (60.9)	39 (24.2)	<0.0001
Reduction in abdominal swelling after having a bowel movement (n, %)	N = 50	N = 39	
	37 (74.0)	28 (71.8)	NS
Self-reported association between intestinal disorders and a change in usual bowel moments (n, %)	N = 56	N = 69	
	42 (75.0)	39 (56.5)	0.049
Type of modification in usual bowel movements (n, %)	N = 42	N = 39	
1. Diarrhea	10 (23.8)	5 (12.8)	NS
2. Constipation	13 (31.0)	17 (43.6)	NS
3. Mixed bowel movements	11 (26.2)	8 (20.5)	NS
4. Other	8 (19.0)	9 (23.1)	NS
Association between intestinal disorders and episodes of psychophysical stress (n, %)	N = 56	N = 69	
	42 (75.0)	39 (56.5)	0.049
Associated comorbidities (n, %)			
1. Anxiety	45 (54.9)	51 (31.7)	0.0008
2. Depression	19 (23.2)	24 (14.9)	NS
3. Bipolar disorder	2 (2.4)	0 (0.0)	NS
4. Schizophrenia	0 (0.0)	0 (0.0)	NS
5. Thyroid diseases	27 (32.9)	15 (9.3)	<0.0001
6. Diabetes mellitus	5 (6.1)	161 (100.0)	<0.0001
7. Pernicious anemia (vitamin B12 deficiency)	2 (2.4)	3 (1.9)	NS
8. Chronic fatigue	27 (32.9)	48 (29.8)	NS
9. Fibromyalgia	24 (29.3)	0 (0.0)	<0.0001
10. Chronic intestinal inflammatory diseases	0 (0.0)	0 (0.0)	NS
11. Chronic headache	16 (19.5)	9 (5.6)	0.001
12. Irritable bowel syndrome	33 (40.2)	9 (5.6)	<0.0001
13. Celiac disease	0 (0.0)	0 (0.0)	NS
14. Gastroesophageal reflux	42 (51.2)	66 (41.0)	NS
15. Other autoimmune diseases	19 (23.1)	0 (0.0)	<0.0001
16. Other non-autoimmune diseases	8 (9.8)	81 (50.3)	<0.0001

NS = not significant; pSS = primary Sjögren’s Syndrome.

**Table 3 nutrients-17-03172-t003:** Questionnaire on self-reported NCWS in pSS patients and controls.

	pSS (n = 82)	Controls (n = 161)	*p*
Worsening of symptoms related to pSS after eating wheat-containing foods (n, %)	18 (21.9)	NA	NA
Clinical manifestations worsened following wheat-containing food intake (n, %)	N = 18		
1. Xerostomia	8 (44.4)		
2. Xerophthalmia	2 (11.2)	NA	NA
3. Xerostomia and xerophthalmia	8 (44.4)		
Degree of symptom worsening (n, %)	N = 18		
1. Mild	6 (33.3)		
2. Moderate	9 (50.0)	NA	NA
3. Severe	3 (16.7)		
Improvement in one or more of the symptoms related to pSS by eliminating wheat-containing foods from the diet (n, %)	N = 18		
	11 (61.1)	NA	NA
Clinical manifestations improved following wheat-containing food elimination (n, %)	N = 11		
1. Xerostomia	3 (27.3)		
2. Xerophthalmia	1 (9.1)	NA	NA
3. Xerostomia and xerophthalmia	7 (63.6)		
Degree of symptom improvement (n, %)	N = 11		
1. Very slight improvement	0 (0.0)		
2. Slight improvement	2 (18.2)		
3. Satisfactory improvement	5 (45.4)	NA	NA
4. Clear improvement which permitted a reduction in therapy	4 (36.4)		
5. Very clear improvement which led to the disappearance of symptoms and permitted therapy to be suspended	0 (0.0)		
Symptoms unrelated to pSS after eating wheat-containing products (n, %)	39 (47.6)	30 (18.6)	<0.0001
Symptoms referred after eating wheat-containing products (n, %)	N = 39	N = 30	
1. Intestinal bloating	28 (71.8)	18 (60.0)	NS
2. Abdominal pain	26 (66.7)	8 (26.7)	0.002
3. Abdominal heaviness	23 (59.0)	15 (50.0)	NS
4. Diarrhea	14 (35.9)	3 (10.0)	0.028
5. Constipation	20 (51.3)	9 (30.0)	NS
6. Asthenia	19 (48.7)	6 (20.0)	0.027
7. Belching	10 (25.6)	6 (20.0)	NS
8. Flatulence	20 (51.3)	1 (3.3)	<0.0001
9. Nausea and/or vomiting	12 (30.8)	9 (30.0)	NS
10. Headache	13 (33.3)	1 (3.3)	0.006
11. Poor motor coordination	7 (17.9)	0 (0.0)	NS
12. Numbness or prickling sensation on the skin	9 (23.1)	1 (3.3)	0.049
13. Anemia	6 (15.4)	0 (0.0)	NS
14. Redness of the skin	3 (7.7)	0 (0.0)	NS
15. Joint pain	18 (46.2)	6 (20.0)	0.045
16. Other (indicate)	0 (0.0)	0 (0.0)	NS
Frequency of symptoms after eating wheat-containing foods (n, %)	N = 39	N = 30	
1. Always	16 (41.0)	0 (0.0)	<0.0001
2. Often (≥3 days/week)	11 (28.2)	24 (80.0)	<0.0001
3. A few days a week (<3 days/week)	6 (15.4)	0 (0.0)	NS
4. A few times a month (once or more a month)	5 (12.8)	6 (20.0)	NS
5. A few times a year (less than once a month)	1 (2.6)	0 (0.0)	NS
6. Less than once a year	0 (0.0)	0 (0.0)	NS
Timing of symptom onset after wheat-containing food intake (n, %)	N = 39	N = 30	
1. Almost immediately (less than an hour)	16 (41.0)	9 (30.0)	NS
2. 1–6 h	21 (53.8)	18 (60.0)	NS
3. 6–24 h	0 (0.0)	3 (10.0)	NS
4. The next day	1 (2.6)	0 (0.0)	NS
5. A few days	1 (2.6)	0 (0.0)	NS
Duration of symptoms (n, %)	N = 39	N = 30	
1. A few minutes	4 (10.3)	12 (40.0)	0.009
2. A few hours	21 (53.8)	15 (50.0)	NS
3. A few days	11 (28.2)	3 (10.0)	NS
4. A few weeks	2 (5.1)	0 (0.0)	NS
5. A few months	1 (2.6)	0 (0.0)	NS
Wheat-containing foods associated with symptom onset (n, %)	N = 39	N = 30	
1. Cereals	5 (12.8)	0 (0.0)	NS
2. Bread	20 (51.3)	15 (50.0)	NS
3. Pasta	26 (66.7)	3 (10.0)	<0.0001
4. Pizza	19 (48.7)	15 (50.0)	NS
5. Biscuits	8 (20.5)	0 (0.0)	0.008
6. Sweets/Candies	12 (30.8)	0 (0.0)	0.001
7. Other (specify)	0 (0.0)	0 (0.0)	NS
Period elapsed since the first episodes of wheat intolerance (months) [Median (IQR)]	120 (48–217)	5.5 (2–63)	0.0001
Subjects who had consulted a doctor/dietician/health care specialist for wheat-intake-related disorders (n, %)	N = 39	N = 30	
	26 (66.7)	18 (60.0)	NS
Professional consulted (multiple answers possible) (n, %)	N = 26	N = 18	
1. Gastroenterologist	19 (73.1)	6 (33.3)	0.021
2. General practitioner	12 (46.2)	9 (50.0)	NS
3. Dietitian	6 (15.4)	6 (33.3)	NS
4. Allergologist	0 (0.0)	18 (100.0)	<0.0001
5. Hematologist	1 (2.6)	0 (0.0)	NS
6. Rheumatologist	1 (2.6)	0 (0.0)	NS
Testing carried out (n, %)	N = 26	N = 18	
1. Celiac disease serology	16 (61.5)	3 (16.7)	0.008
2. Prick test for food allergy	7 (17.9)	14 (77.8)	0.003
3. Esophagogastroduodenoscopy	16 (61.5)	3 (16.7)	0.008
4. Abdomen ultrasound examination	1 (3.8)	3 (16.7)	NS
5. Patch test for nickel allergy	1 (3.8)	4 (22.2)	NS
6. Genetic (HLA) typing	1 (3.8)	0 (0.0)	NS
Possible explanation for referred symptoms given by specialists (n, %)	N = 26	N = 18	
1. Celiac disease	8 (30.8)	3 (16.7)	NS
2. Wheat allergy	0 (0.0)	0 (0.0)	NS
3. Irritable bowel syndrome	9 (34.6)	1 (5.6)	NS
4. NCWS	3 (11.5)	0 (0.0)	NS
5. Gastritis	1 (3.8)	0 (0.0)	NS
6. Diverticulosis	1 (3.8)	3 (16.7)	NS
7. No explanation	5 (19.2)	12 (66.7)	0.004
Subjects who had undergone a period of wheat-containing product elimination	N = 39	N = 30	
(n, %)	20 (51.3)	0 (0.0)	<0.0001
Improvement in one or more of the symptoms by eliminating wheat-containing foods from the diet (n, %)	N = 20		
	19 (95.0)	NA	NA
Subjects following a wheat-free diet at the time of recruitment (n, %)	N = 20		
	15 (75.0)	NA	NA
Intake of products containing ancient grains during wheat-free diet (n, %)	N = 20		
	10 (50.0)	NA	NA
Lack of symptom onset when taking ancient grains (compared to modern grains) (n, %)	N = 10		
	8 (80.0)	NA	NA

IQR = interquartile range; NA = not applicable; NS = not significant; NCWS = non-celiac wheat sensitivity; pSS = primary Sjögren’s Syndrome; SD = standard deviation.

**Table 4 nutrients-17-03172-t004:** Questionnaire on self-reported intolerance to other non-wheat-containing foods in pSS patients and controls.

	pSS (n = 82)	Controls (n = 161)	*p*
Worsening of the symptoms related to pSS after eating non-wheat-containing foods (n, %)	10 (12.2)	NA	NA
Clinical manifestations worsened following non-wheat-containing food intake (n, %)	N = 10		
1. Xerostomia	5 (50.0%)		
2. Xerophthalmia	0 (0.0)	NA	NA
3. Xerostomia and xerophthalmia	5 (50.0%)		
Degree of symptom worsening (n, %)	N = 10		
1. Mild	1 (10.0)		
2. Moderate	7 (70.0)	NA	NA
3. Severe	2 (20.0)		
Improvement in one or more of the symptoms related to pSS by eliminating non-wheat-containing foods from the diet (n, %)	N = 10		
	8 (80.0)	NA	NA
Clinical manifestations improved following non-wheat-containing food elimination (n, %)	N = 8		
1. Xerostomia	0 (0.0)		
2. Xerophthalmia	3 (37.5)	NA	NA
3. Xerostomia and xerophthalmia	5 (62.5)		
Degree of symptom improvement (n, %)	N = 8		
1. Very slight improvement	0 (0.0)		
2. Slight improvement	3 (37.5)		
3. Satisfactory improvement	1 (12.5)	NA	NA
4. Clear improvement permitting a reduction in therapy	4 (50.0)		
5. Very clear improvement which led to the disappearance of symptoms and permitted therapy to be suspended	0 (0.0)		
Symptoms unrelated to pSS after eating non-wheat-containing products (n, %)	35 (42.7)	21 (13.0)	<0.0001
Symptoms referred after eating non-wheat-containing products (n, %)	N = 35	N = 21	
1. Intestinal bloating	19 (54.3)	9 (42.9)	NS
2. Abdominal pain	18 (51.4)	6 (28.6)	NS
3. Abdominal heaviness	18 (51.4)	6 (28.6)	NS
4. Diarrhea	14 (40.0)	1 (4.8)	0.01
5. Constipation	11 (31.4)	12 (57.1)	NS
6. Asthenia	12 (34.2)	3 (14.3)	NS
7. Belching	12 (34.2)	1 (4.8)	0.027
8. Flatulence	19 (54.3)	3 (14.3)	0.007
9. Nausea and/or vomiting	12 (34.3)	6 (28.6)	NS
10. Headache	9 (25.7)	3 (14.3)	NS
11. Poor motor coordination	5 (14.3)	0 (0.0)	NS
12. Numbness or prickling sensation on the skin	6 (17.1)	0 (0.0)	NS
13. Anemia	3 (8.6)	0 (0.0)	NS
14. Redness of the skin	6 (17.1)	0 (0.0)	NS
15. Joint pain	7 (20.0)	0 (0.0)	NS
16. Other (specify)	0 (0.0)	0 (0.0)	NS
Frequency of symptoms after eating non-wheat-containing foods (n, %)	N = 35	N = 21	
1. Always	14 (40.0)	15 (71.4)	0.045
2. Often (≥3 days/week)	14 (40.0)	6 (28.6)	NS
3. A few days a week (<3 days/week)	2 (5.7)	0 (0.0)	NS
4. A few times a month (once or more a month)	4 (11.4)	0 (0.0)	NS
5. A few times a year (less than once a month)	1 (2.9)	0 (0.0)	NS
6. Less than once a year	0 (0.0)	0 (0.0)	NS
Timing of symptom onset after non-wheat-containing food intake (n, %)	N = 35	N = 21	
1. Almost immediately (less than an hour)	14 (40.0)	6 (28.6)	NS
2. 1–6 h	18 (51.4)	12 (57.1)	NS
3. 6–24 h	3 (8.6)	0 (0,0)	NS
4. The next day	0 (0.0)	3 (14.3)	NS
5. A few days	0 (0.0)	0 (0.0)	NS
Duration of symptoms (n, %)	N = 35	N = 21	
1. A few minutes	7 (20.0)	6 (28.6)	NS
2. A few hours	22 (62.9)	15 (71.4)	NS
3. A few days	6 (17.1)	0 (0.0)	NS
4. A few weeks	0 (0.0)	0 (0.0)	NS
5. A few months	0 (0.0)	0 (0.0)	NS
Period elapsed since the first episodes of wheat intolerance (months) [Median (IQR)]	129 (60–240)	120 (36–240)	NS
Subjects who had consulted a doctor/dietician/health care specialist for wheat-intake-related disorders (n, %)	N = 35	N = 21	
	28 (80.0)	3 (14.3)	<0.0001
Professional consulted (multiple answers possible) (n, %)	N = 28	N = 3	
1. Gastroenterologist	12 (42.9)	2 (66.7)	NS
2. General practitioner	17 (60.7)	2 (66.7)	NS
3. Dietitian	3 (10.7)	0 (0.0)	NS
4. Allergologist	2 (7.1)	0 (0.0)	NS
5. Hematologist	1 (3.6)	0 (0.0)	NS
6. Rheumatologist	1 (3.6)	0 (0.0)	NS
Testing carried out (n, %)	N = 28	N = 3	
1. Patch test for nickel allergy	6 (21.4)	1 (33.3)	NS
2. Prick test for food allergy	9 (32.1)	1 (33.3)	NS
3. Lactose breath test	11 (39.3)	1 (33.3)	NS
4. Esophagogastroduodenoscopy	5 (17.9)	2 (66.7)	NS
5. Search for food-specific IgE	1 (3.6)	0 (0.0)	NS
Possible explanation for referred symptoms given by specialists (n, %)	N = 28	N = 3	
1. Nickel allergy	4 (14.3)	0 (0.0)	NS
2. Food allergy	5 (17.9)	0 (0.0)	NS
3. Lactose intolerance	11 (39.3)	1 (33.3)	NS
4. Irritable bowel syndrome	10 (35.7)	2 (66.7)	NS
5. No explanation	6 (21.4)	1 (33.3)	NS

IgE = Immunoglobulin E; IQR = interquartile range; NA = not applicable; NS = not significant; pSS = primary Sjögren’s Syndrome.

**Table 5 nutrients-17-03172-t005:** Multiple logistic regression analysis analyzing association between pSS and self-reported NCWS as well as SRMI and MFS.

	OR	CI 95%	*p*
Self-reported NCWS	3.96	2.2–7.13	<0.0001
SRMI	6.99	3.07–15.93	<0.0001
MFS	4.27	2.1–8.68	0.0017

MFS = multiple food sensitivity; NCWS = non-celiac wheat sensitivity; pSS = primary Sjögren’s Syndrome; SRMI = self-reported milk intolerance.

## Data Availability

The data presented in this study are available on request from the corresponding author due to being a part of an ongoing study.

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
