# Peer review of "Self-Reported Non-Celiac Wheat Sensitivity and Other Food Sensitivities in Patients with Primary Sjögren’s Syndrome"

_nutrients, 2025, doi:10.3390/nu17193172_

Round 1

Reviewer 1 Report

Comments and Suggestions for Authors

Authors present an interseting study, generally well planned and executed. They clearly stated most important limitations. What could be possibly checked in the future requires gastroscopy with biopsy - CD patients usually accumulate high count of gamma delta T cells within duodenal epithelium, it would be interesting to check if the same could be noted in those pSS patients that report some sensitivity to gluten/wheat. Moreover, similar assessment in the peripheral blood could be of interest. 

As for the current study I have only two suggestions. Both could be included and I would strongly suggest to do it. If the results are not significant, those results could be kept in supplement.
How does this correlate with ESSDAI/ESSPRI? Did authors try to subdivide pSS patients into low vs moderate/high value groups and compare the perceived intolerance? 
What treatment did the patients receive? Did it differ between those with or without perceived intolerance?

Author Response

Authors present an interesting study, generally well planned and executed. They clearly stated most important limitations.

R: We thank the reviewer for his appreciation of our paper. He clearly understood its intent, including the analysis of the obvious limitations of this preliminary study.

What could be possibly checked in the future requires gastroscopy with biopsy -CD patients usually accumulate high count of gamma delta T cells within duodenal epithelium, it would be interesting to check if the same could be noted in those pSS patients that report some sensitivity to gluten/wheat. Moreover, similar assessment in the peripheral blood could be of interest. 

R: We thank the reviewer for the extremely useful observation, and we will certainly attempt to accurately evaluate the immune response and inflammatory features in patients with pSS and NCWS in the near future. Specifically, in the prospective study we are already conducting, due to the difficulties in proposing and performing duodenal and/or rectal endoscopic biopsies before and after the start of a rigorous WFD in patients with pSS, we planned to study the following markers:

1) inflammatory markers in peripheral blood samples using ELISA methods (IL-6, IL-10, IL-18, IL-21, TNF-α, etc.)

2) intestinal permeability markers in peripheral blood samples using ELISA methods (zonulin, IFABP, f-actin, etc.)

3) lymphocyte expression patterns in peripheral blood samples using flow cytometry (Th1, Th17, Th22, Treg, NK, Tgamma/delta, etc.)

As for the current study I have only two suggestions. Both could be included and I would strongly suggest to do it. If the results are not significant, those results could be kept in supplement.
How does this correlate with ESSDAI/ESSPRI? Did authors try to subdivide pSS patients into low vs moderate/high value groups and compare the perceived intolerance? 

R: We thank the reviewer for his very important comment. We had forgotten to include in the supplementary tables the divisions into disease activity classes according to the ESSDAI score. Similarly, in the original version of main text we have reported only:

-Results: “None of the demographic, clinical and immunological features typical of pSS were significantly different between pSS patients with (n=39) or without (n=43) self-reported NCWS,...”

-Discussion: “Analysis of the clinical and immunological features of our pSS study population showed that the disease was under good control in most of the patients enrolled, with ESSDAI and ESSPRI values ​​classifying them at a low, or at most moderate degree of disease activity, both locally and systemically.' and 'As regards the secondary outcome of our research, none of the demographic, clinical (and immunological features analyzed showed statistically significant differences between the 'wheat sensitive' and 'non-wheat sensitive' pSS subjects, …”.

Thus, to better explain the lack of any difference in the clinical presentation of pSS patients, according to the presence or absence of self-reported NCWS, we have now added in Supplemental Tables 1 and 4 (yellow underlined) the subdivision into individual disease activity classes according to ESSDAI score, and we have modified one of the sentences in the discussion as follows (yellow underlined): “As regards the secondary outcome of our research, none of the demographic, clinical (including the ESSPRI and the ESSDAI disease activity score), immunological and therapeutic features analyzed showed statistically significant differences between the ‘wheat sensitive’ and ‘non-wheat sensitive’ pSS subjects, with the exception of SRMI and MFS, which had a higher prevalence in patients who self-reported NCWS compared to those who did not”. Ù

In addition, we added, in supplemental table 1, also number and percentage of pSS patients with xerostomia and/or xerophthalmia.

What treatment did the patients receive? Did it differ between those with or without perceived intolerance?

R: We thank the reviewer for this valuable input. As evidenced by the ESSDAI scores, most pSS patients had low or moderate disease activity and symptom intensity. None of the patients included in our study were receiving systemic steroids or immunosuppressive drugs. Furthermore, the use of these drugs in those patients who underwent EGD to rule out CeD would have automatically excluded them from the study (we modified the main text to include the use of immunosuppressive drugs among the exclusion criteria in patients undergoing EGD, as we have forgotten to specify this in the first submission).

The treatments used by the patients were limited to the following:

- Patients with xerostomia: Most enrolled patients used mechanical or gustatory stimulation techniques (28 in the self-reported NCWS group and 30 in the non-self-reported NCWS group), while 11 used artificial saliva (6 in the self-reported NCWS group and 5 in the non-self-reported NCWS group).

- Patients with xerophthalmia: lubricant ointments and topical cyclosporine. Specifically, only 6 patients used topical cyclosporine (2 in the self-reported NCWS group and 4 in the non-self-reported NCWS group), while all others used lubricant ointments (34 in both self-reported and non self-reported NCWS group).

We have modified Supplemental Tables 1 and 4 (yellow underlined) to include the above reported information. In addition, we have modified the main text in Results and Discussion as follows (yellow underlined):

-Results: “None of the demographic, clinical, immunological and therapeutic features typical of pSS were significantly different between pSS patients with (n=39) or without (n=43) self-reported NCWS, although the frequency of both SRMI (39.5% vs 11.9%, P=0.01) and MFS (65.7% vs 23.8%; p=0.0004) was higher in the former (Supplemental Tables 3-4)”.

-Discussion: “As regards the secondary outcome of our research, none of the demographic, clinical (including the ESSPRI and the ESSDAI disease activity score), immunological and therapeutic features analyzed showed statistically significant differences between the ‘wheat sensitive’ and ‘non-wheat sensitive’ pSS subjects, with the exception of SRMI and MFS, which had a higher prevalence in patients who self-reported NCWS compared to those who did not.”

Reviewer 2 Report

Comments and Suggestions for Authors

This manuscript is a good/necessary/timely addition to the scientific literature pertaining to  primary Sjögren's syndrome.

The study is well designed and executed. However, although I would posit the following points to the authors and do so in the spirit of helpfullness and certainly not as a "damning critique"; the points raised, will I hope, not only improve the manuscript (from a scientific perspective) but also be helpful to the "general reader" who is not a "specialist" in the subject matter under scrutiny.

  1. Should there not be a single sentence, in the "Introduction" in relation to: Secondary Sjögren's Syndrome? I am aware that the foregoing is a "hotly debated " subject.
  2. Again, briefly, in the "Introduction" would it not help the "general reader" if the different stages of the syndrome were to be mentioned?
  3. Ditto, the gender disparity of the syndrome.
  4. Ditto, reports of the syndrome being reported many years ago (1892-193?); despite the fact that there is still no "outright cure".
  5. Might it be useful to include/allude to the following two references (?): 

    (i) https://www.ncbi.nlm.nih.gov/books/NBK431049/

    (ii) https://www.ihi.europa.eu/news-events/newsroom/new-subgroups-sjogrens-patients-mean-better-treatments-possible

  6.  

    Ditto, in the discussion, the possibilities of  the use of  (i)  low-dose IL-2 therapy,

    (ii) nipocalimab and (iii) ianalumab to help to treat the "syndrome" in the, not too distant, future?

Author Response

This manuscript is a good/necessary/timely addition to the scientific literature pertaining to primary Sjögren's syndrome. The study is well designed and executed. However, although I would posit the following points to the authors and do so in the spirit of helpfullness and certainly not as a "damning critique", the points raised, will I hope, not only improve the manuscript (from a scientific perspective) but also be helpful to the "general reader" who is not a "specialist" in the subject matter under scrutiny.

R: We thank the reviewer for his appreciation of our work. We would like to point out that any constructive criticism that not only enhances our work's scientific strength but also makes it more understandable for a non-specialist audience is most welcome. Therefore, we thank the reviewer for the important suggestions. In our original paper we did not report many details about SS due to ‘brevity issues’, as we didn't want to overwhelm what we felt was already a long text. However, the reviewer is absolutely right, and a non-rheumatologist reader might be confused due to the lack of some ‘general information’. Thus, to better introduce our paper, we've added the following sentences, which you can find underlined in green throughout the text.

  1. Should there not be a single sentence, in the "Introduction" in relation to: Secondary Sjögren's Syndrome? I am aware that the foregoing is a "hotly debated" subject.

R: We have added in the Introduction the following sentence: ‘‘It could be defined as primary Sjögren's Syndrome (pSS), if not associate with other autoimmune disorders, or as secondary, if it is diagnosed together with other systemic rheumatological diseases (e.g. systemic lupus erythematosus, systemic sclerosis or rheumatoid arthritis). Nevertheless, there is also a third option in which Sjögren's Syndrome is described neither alone nor in the context of other systemic autoimmune disorders, but in association with organ-specific autoimmune diseases, such as celiac disease (CeD), primary biliary cholangitis, autoimmune hepatitis, autoimmune thyroid diseases, etc. [6–8]”

  1. Again, briefly, in the "Introduction" would it not help the "general reader" if the different stages of the syndrome were to be mentioned?

R: We have added the following sentence: ‘‘Clinically, pSS is characterized primarily by ocular and/or mouth dryness (i.e. ‘sicca syndrome'). At least one of these symptoms affects 98% of pSS patients, with both occurring in 90% [5–7]. However, the disease generally progresses slowly over time [8,17]: an initial pre-clinical phase, characterized only by antibody positivity in the absence of clear signs and symptoms (pre-disease); a phase characterized only by glandular involvement with xerostomia and xerophthalmia (stage 1); a phase characterized by systemic extraglandular manifestations, including fatigue, confusion, but also neurological, musculoskeletal, nephrological, skin, and other organs involvement (stage 2); a phase with a possible, though infrequent (about 5% [18]), evolution towards a lymphoid malignancy, usually non-Hodgkin's lymphoma (stage 3)”.

  1. Ditto, the gender disparity of the syndrome.

R: We have added the following sentence: ‘‘mainly affecting middle-aged (sixth and seventh decade of life) women (female-to-male ratio: 9/1) [3–7]”.

  1. Ditto, reports of the syndrome being reported many years ago (1892-193?); despite the fact that there is still no "outright cure".

R: We have added the following sentences:

‘‘Since the first report of a ‘sicca syndrome’ in the early 1900s, by the Swedish physician Henrik Sjögren, and despite several physiopathological and clinical advances in understanding this condition have been performed [6,7,13]”…

“In addition, some authors have identified different subgroups of patients, characterized by various degrees of glandular and/or extraglandular inflammation and symptoms, which probably could benefit of differentiate therapeutic approaches [14]”.

  1. Might it be useful to include/allude to the following two references (?): 

(i) https://www.ncbi.nlm.nih.gov/books/NBK431049/

(ii) https://www.ihi.europa.eu/news-events/newsroom/new-subgroups-sjogrens-patients-mean-better-treatments-possible

R: We thank the reviewer for the suggestion. We have added these references together with some others (green underlined).

  1. Ditto, in the discussion, the possibilities of the use of (i) low-dose IL-2 therapy, (ii) nipocalimab and (iii) ianalumab to help to treat the "syndrome" in the, not too distant, future?

R: We thank the reviewer for this important insight. We have added this short paragraph focusing on the actual and future prospectives of pSS treatment at the beginning of ‘Discussion’:

“The clinical management of patients with pSS has not yet found unanimous consensus in the scientific community [12,15,16], likely because this condition presents clusters of patients with different inflammatory/immune and clinical features [14]. To date, therapy is primarily based on the use of lubricant ointments, local immunosuppressants, artificial saliva, and, in cases of severe extraglandular involvement, systemic immunosuppressants, such as glucocorticoids, methotrexate, azathioprine, hydroxychloroquine, etc., and sometimes, even the use of anti-CD20 B lymphocyte-depleting agents (i.e., rituximab) [7,47]. More recently, new therapeutic perspectives have been applied in patients affected by pSS, including low doses of IL-2 (able to modulate immune cells response enhancing Tregs and suppressing the pro-inflammatory subsets) [48,49], nipocalimab (a monoclonal antibody that reduces circulating IgG levels by selectively blocking their interactions with the neonatal Fc receptor) [50,51], and ianalumab (a B cell-activating factor receptor inhibitor and B-cell depleter monoclonal antibody) [52]”.

Round 2

Reviewer 1 Report

Comments and Suggestions for Authors

Authors have sufficiently addressed my comments.